# Mechanical Behaviors of Existing Large-Diameter Tunnel Induced by Horseshoe-Shaped Undercrossing Twin Tunnels in Gravel

**Jianye Li [1,2], Qian Fang [1,2,*], Xiang Liu [3], Jianming Du [1,2], Gan Wang [1,2] and Jun Wang [1,2]**

1   Key Laboratory of Urban Underground Engineering of Ministry of Education, Beijing Jiaotong University, Beijing 100044, China; 20115024@bjtu.edu.cn (J.L.); 19115013@bjtu.edu.cn (J.D.); 21115015@bjtu.edu.cn (G.W.); 17724801835@163.com (J.W.)
2   School of Civil Engineering, Beijing Jiaotong University, Beijing 100044, China
3   Liaoning Key Laboratory of Marine Environmental Bridge and Tunnel Engineering, Dalian Maritime University, Dalian 116026, China; xliu@dlmu.edu.cn
*   Correspondence: qfang@bjtu.edu.cn; Tel.: +86-10-5168-8115

**Abstract:** This article investigates and presents a case study on the Beijing Subway Line 12 excavation beneath the existing Qinghuayuan Tunnel. The composite pre-reinforcement technique was used in conjunction with the shallow tunneling method to control the distortion of the existing large-diameter tunnel. When building twin tunnels underneath, this strategy considerably decreased the impact on the existing large-diameter tunnel. To systematically study the mechanical response of the existing large-diameter tunnel, a variety of sensors was embedded in the prefabricated segments just above the new twin tunnels. During the undercrossing twin tunnels procedure, the earth pressure, tunnel crown settlement, opening width of the segment joint, and the circumferential strain of the large-diameter existing tunnel were all measured. The settlement development of the existing large-diameter tunnel was categorized under six stages: (1) sedimentation, (2) heave, (3) second sedimentation, (4) second heave, (5) third sedimentation, and (6) steady state. The joint opening of the existing large-diameter tunnel changed sharply during the new undercrossing twin tunnels. The earth pressure and concrete stress of the linings rapidly increased during the new undercrossing twin tunnels. The majority of the reinforcement and concrete stresses were compressive and far lower than the yield strength, indicating that the tunnel was in a safe working condition.

**Keywords:** large-diameter tunnel; mechanical behaviors; undercrossing; composite pre-reinforcement; shallow tunneling method

## 1. Introduction

With the increased use of subway lines in highly populated metropolitan areas, new tunnels constructed beneath older tunnels are becoming more common [1–7]. The primary difficulties in such parallel tunneling projects are ensuring the safety and serviceability of the current tunnel while building the new tunnel. The mechanical characteristics of the existing tunnel are difficult to investigate, owing to the intrinsic intricacies of soil–tunnel interactions.

Many researchers over the last few decades have evaluated the influence of undercrossing tunneling on existing tunnels using field investigations and monitoring [8–14], analysis methods [14–18], numerical modeling [18–21], and laboratory tests [22–24]. Li and Yuan [8] investigated the reaction of an existing tunnel during new undercrossing twin tunnels. Fang et al. [9] showed the sedimentations of existing shield tunnels and the ground surfaces following the excavation of new twin horseshoe-shaped tunnels. Jin et al. [10] described a method for protecting in-tunnel grouting when building twin tunnels below existing twin tunnels. The sedimentation tank and the produced stress of the existing tunnel induced



by shield tunneling were also analyzed using field monitoring data. Chen et al. [11] investigated the mechanical responses of twin tunnels in sandy soil strata caused by nearer shield undercrossing. Yao et al. [12] used finite element analysis (FEA) and site monitoring to study the deformation response of overlapping tunnels in Tianjin, China, taking into account the features of different tunneling types.

According to the literature, there is a lack of an idea of the reality of the load evolution and mechanical response of the existing large-diameter tunnel during the construction of the new twin horseshoe-shaped tunnels. As a result, this article offers a thorough case analysis of new twin horseshoe-shaped tunnels constructed below Beijing's existing Qinghuayuan Tunnel in gravel. The present tunnel reinforcing measure (composite pre-reinforcement) was described and reported. The settlement, longitudinal joint tensioning, load development process of earth pressures, and hoop stress of existing tunnels were analyzed using monitoring data. The findings of this research may be useful in the design and construction of future tunneling projects.

## 2. Project Overview

### 2.1. The Existing Qinghuayuan Tunnel

The Jing-Zhang High-Speed Railway is a major railway connecting Beijing and Zhangjiakou (350 km/h) (Figure 1). The main line of the Jing-Zhang High-Speed Railway is approximately 174.0 km long. It was used during the 2022 Olympic Winter Games in Beijing [13].

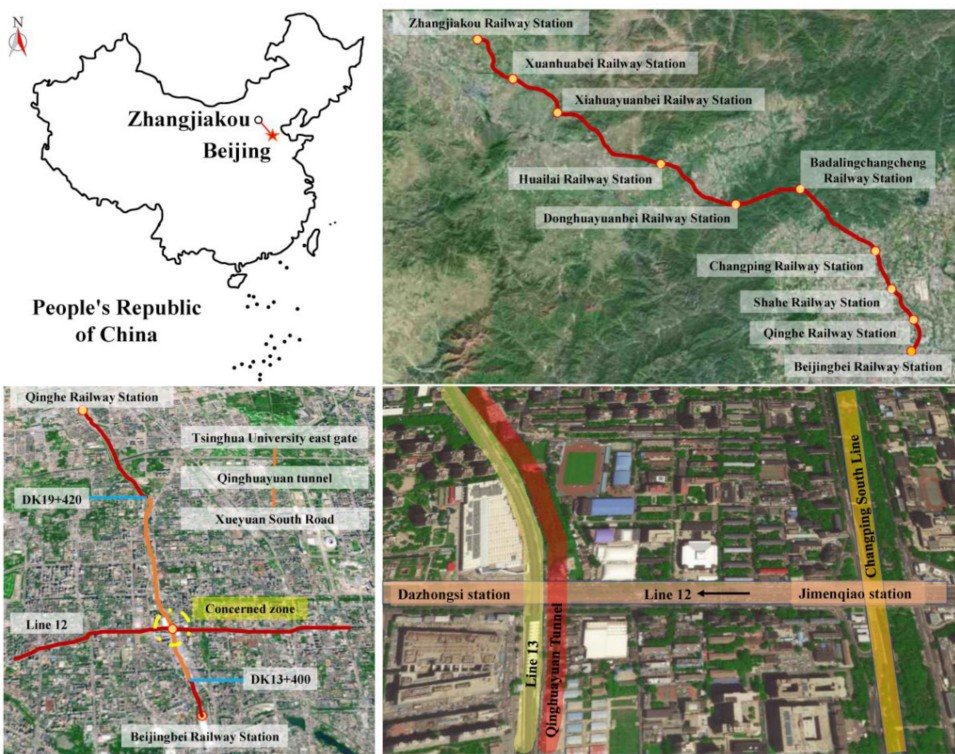

**Figure 1.** Views of Jing-Zhang High-speed Railway, Qinghuayuan Tunnel, and Subway Line 12.

The existing Qinghuayuan Tunnel is a component of the Jing-Zhang High-Speed Railway. It is located in the heavily populated Haidian neighborhood of Beijing. It begins on Xueyuan South Road and terminates at Tsinghua University's East Gate (Figure 1). The maximum inclination of the Qinghuayuan Tunnel is roughly 3%, and the minimum profile curvature radius is 995 m. The tunnel arch's maximum depth of embedment is approximately 29 m. The middle portion of the Qinghuayuan Tunnel, with a length of 4.45 km, was excavated by a slurry pressure balance tunnel boring machine. The existing Qinghuayuan Tunnel is a circle-shaped tunnel. The inner and outer diameters of the tunnel

linings are 11.1 m and 12.2 m, respectively. Each segment has a thickness of 0.55 m and a length of 2 m. The plan of the existing Qinghuayuan Tunnel and Beijing Subway Line 12 is shown in Figure 1. Figure 2 depicts the concerned zone of research.

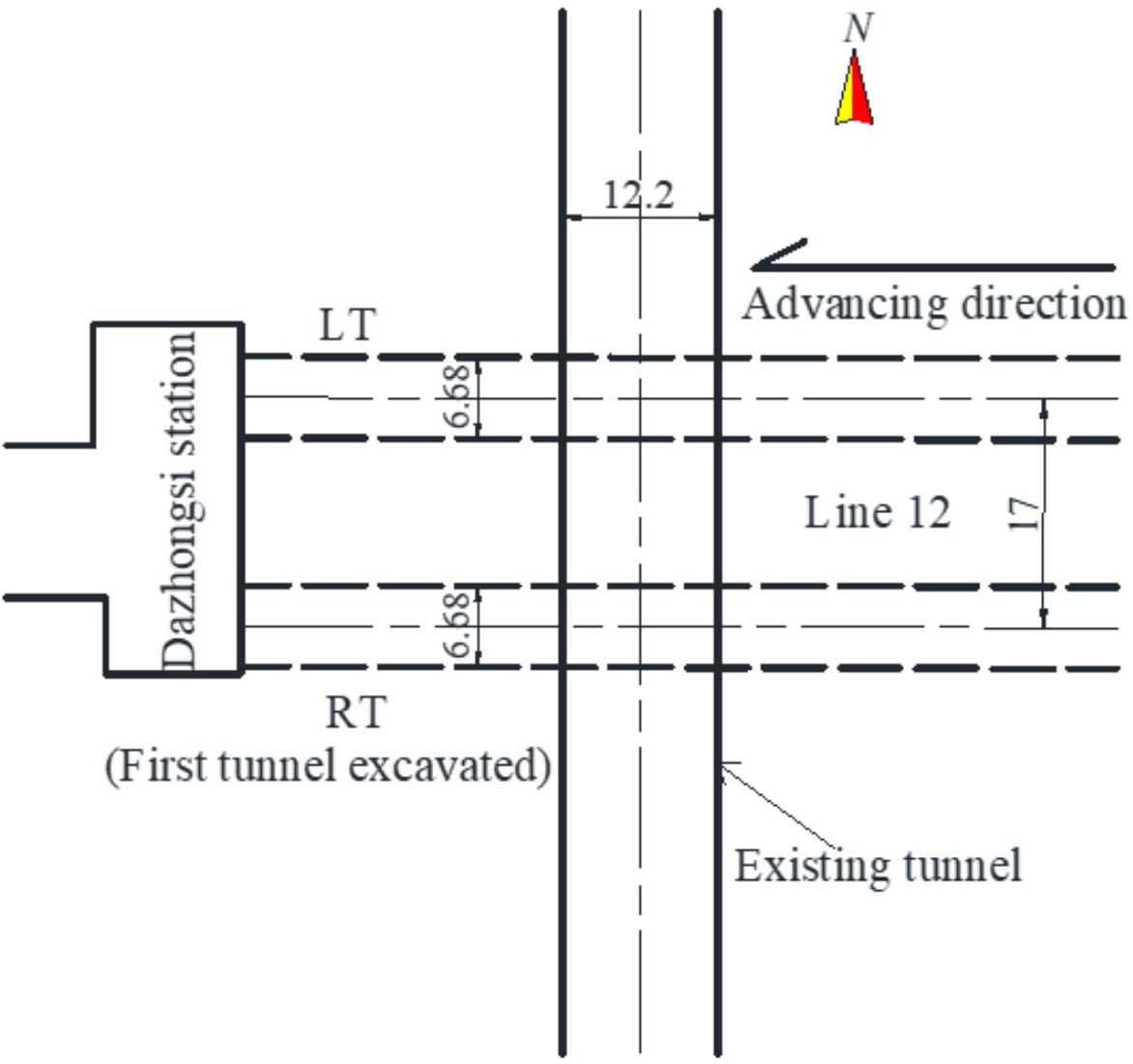

**Figure 2.** Diagram of existing and new tunnels (unit: m).

### 2.2. The New Subway Line 12

Figure 2 depicts the newly constructed east-west horseshoe twin tunnels part of Beijing Subway Line 12. The distance between them is around 18 m. The new twin tunnels are horizontally parallel. They were dug using the shallow tunneling method. The method based on the manual excavation is specifically suited for developing shallow tunnels. The twin tunnels' excavation width and height are 6.68 m and 6.82 m, respectively. The primary and secondary linings have thicknesses of 350 mm and 300 mm, respectively (Figure 3).

### 2.3. Geology and Ground Conditions

Figure 4 displays the project's typical geological profile. Back fill, silty clay, silty sand, and gravel make up the site soil. According to the soil profile, the junction location of the Qinghuayuan Tunnel and Bejing Subway Line 12 is mostly gravel. The groundwater is located 24.0 m below the earth's surface.

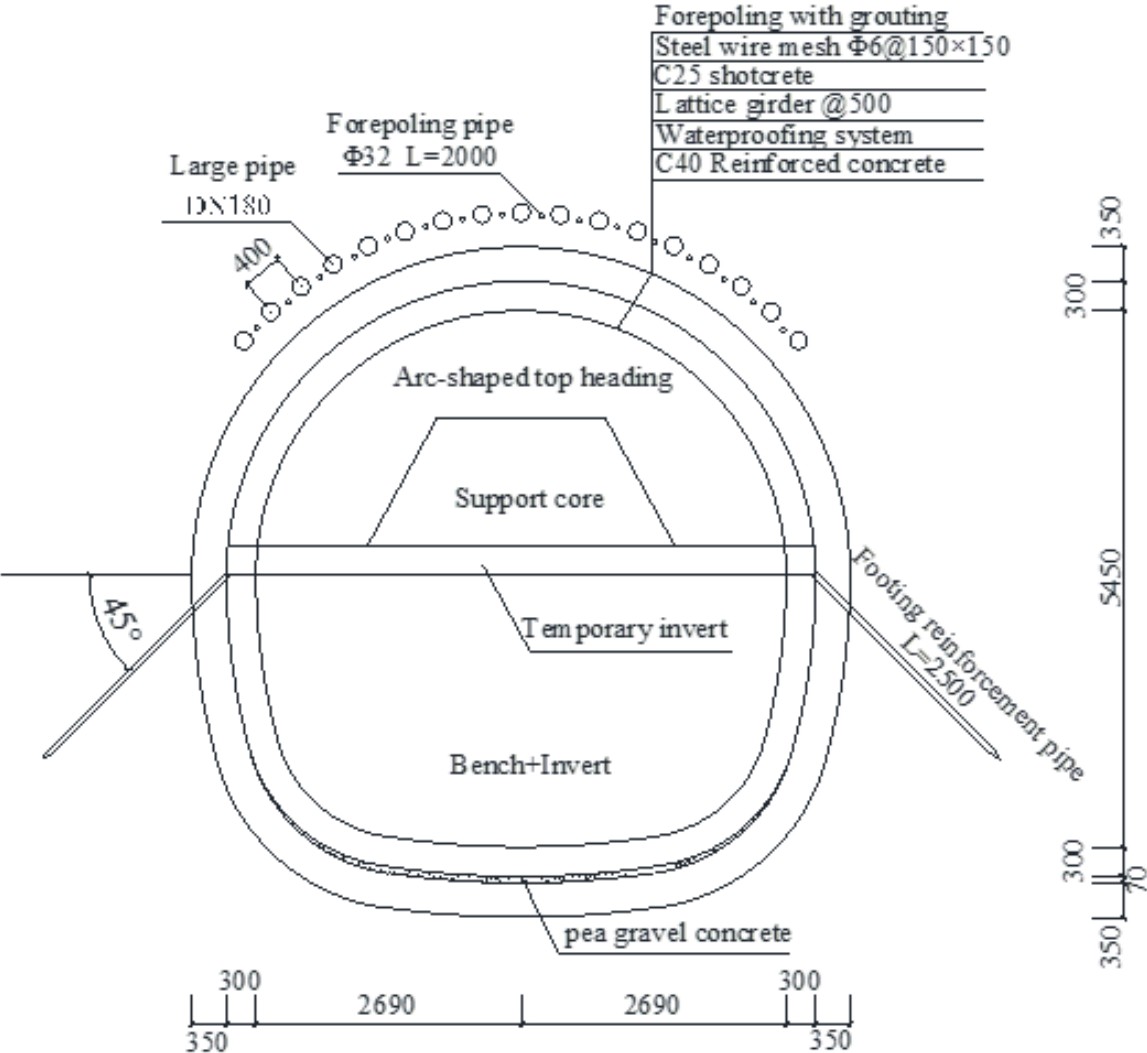

**Figure 3.** Cross-section of the new tunnel (unit: mm).

*2.4. Protection Scheme of the Existing Tunnel*

The connection area of the existing and new tunnels is situated in the water-rich sand and pebble strata. Once the excavation is carried out, the initial equilibrium is easily disrupted, resulting in much deformation due to the loss of support at the hollow surface. Especially when the sand and pebble spall at the top of the underground works, this may lead to the collapse of the strata, causing great difficulties for construction and design. To ensure the security of the current large-diameter shield tunnel, support or pre-reinforcement measures must be taken before the twin tunnels are undercrossing. In this case, the combination reinforcement scheme with a pipe roof and deep hole grouting and the in-tunnel grouting protection method were adopted. Figure 5 depicts the longitudinal view and cross-sections of the combined pre-reinforcement measurements.

Figure 6 depicts a total of 16 grouting holes, each around 35 m long. A sleeve pipe is used to finish the grouting using a blend of conventional Portland cement and sodium silicate slurry. The grout hole had a diameter of 25 mm, and the diffusion radius was supposed to be 0.6 m. The grouting pressure was kept to less than 0.8 MPa. The water-glass slurry has a quick condensation time and an excellent pre-reinforcement effect, whereas the cement slurry takes a lengthy time to set but has a high strength later on. The cement slurry has a long setting time but high strength in the later stages, resulting in a good long-term reinforcement effect. The longitudinal section of the deep hole grouting is lapped by 2 m, and a stop wall is set before the next section of grouting. A 0.3-m thick C25 shotcrete stop

wall is set at the tunnel's surface outside the core soil of the upper step before each section of deep hole grouting, and the core soil is protected by 0.05-m thick C25 shotcrete. The angle of insertion of the grouting holes is controlled within the range of 4–10°. To reduce the settlement of the pipe shed during construction, a self-propelled pipe shed was used with side-welded latches to link the steel pipes together and form an integral support. The tube sheds were made of DN180 hollow steel tubes with a thickness of 12 mm, an insertion angle of 1–3°, and a circular spacing of 300 mm.

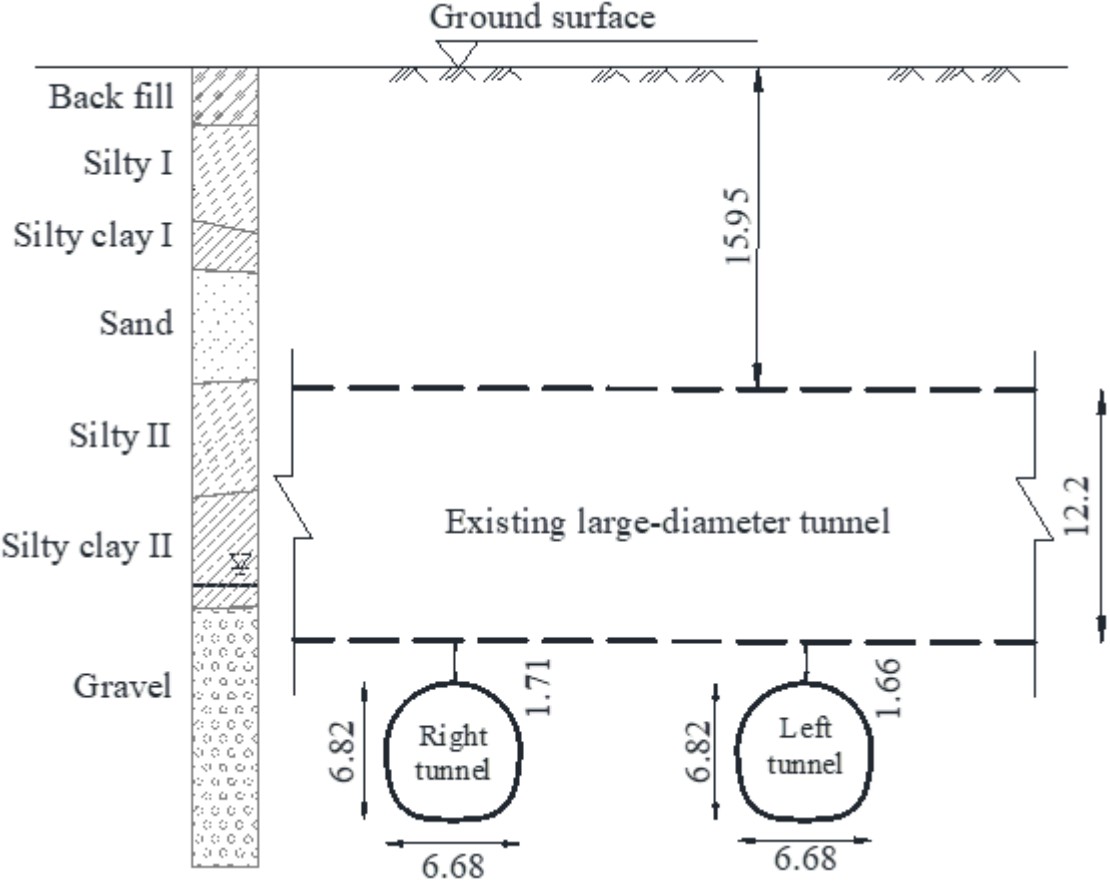

**Figure 4.** Typical intersection zone soil profile (unit: m).

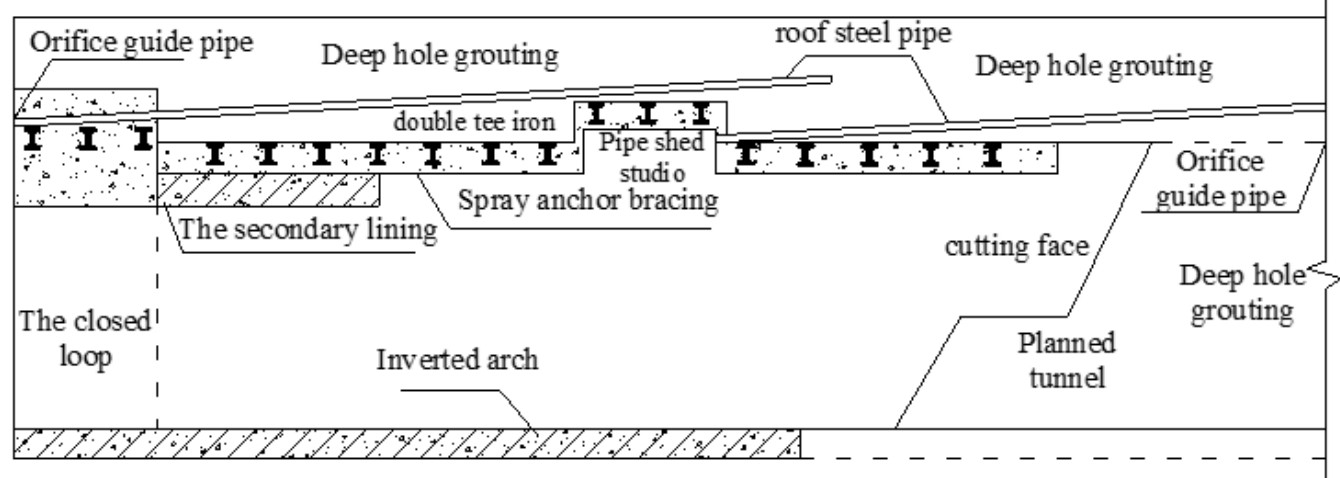

**Figure 5.** Diagram of composite pre-support with overrun pipe shed and deep hole grouting.

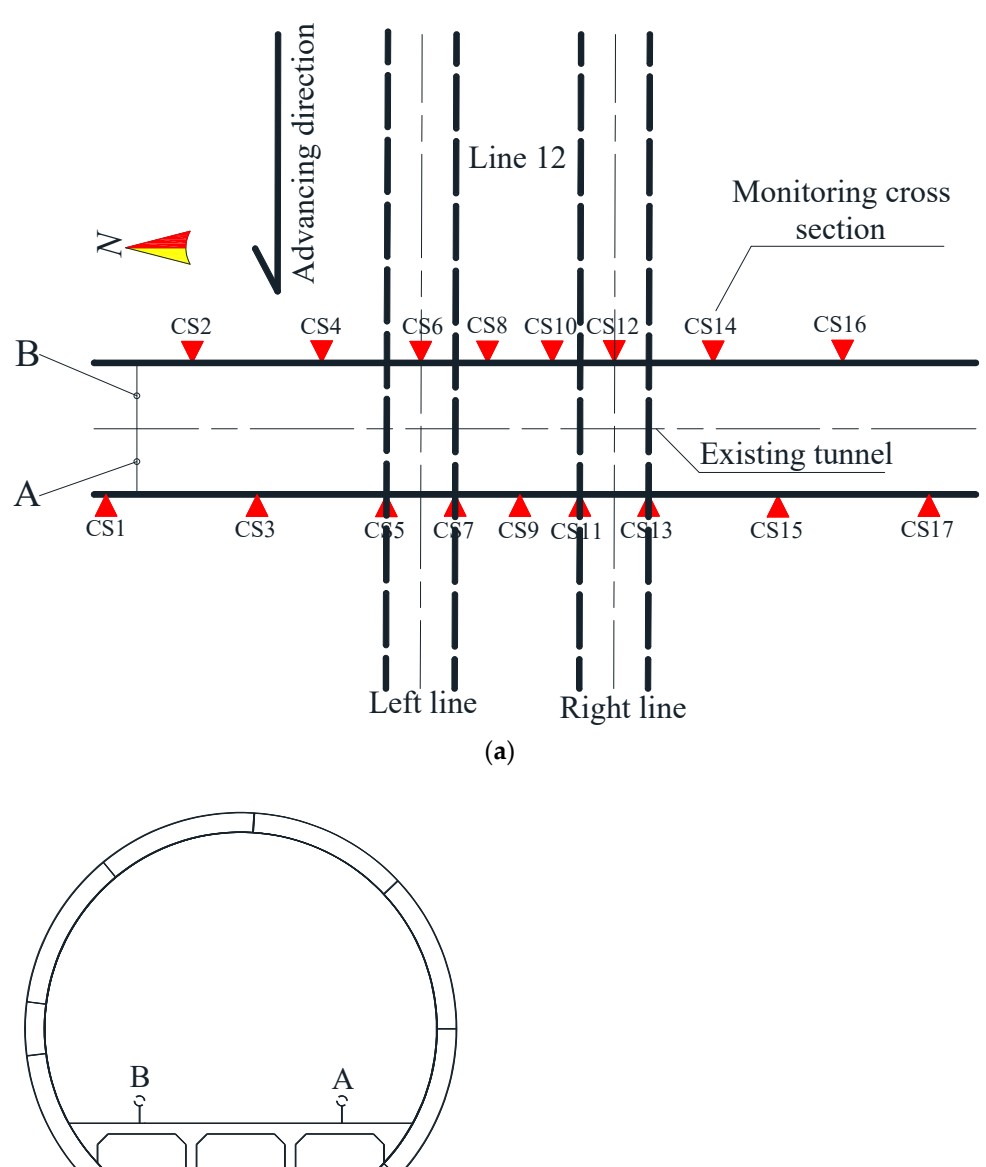

**Figure 6.** (**a**) Plane layout of the monitoring section along the existing large-diameter tunnel and (**b**) layout of the monitoring points.

In addition, the existing tunnel has been reinforced with additional predeposited grout holes (from 10 to 25) with a diameter of 50 mm, and the surrounding soil has been reinforced with grout in advance within 15 m on both sides of the new subway tunnel for a total of 27 rings.

## 3. Monitoring Scheme and Tunnel Construction

### 3.1. Layout of the Monitoring System

The deformation, stress, and joint displacement of the tunnel are continuously monitored according to the mechanical characteristics of large-diameter shield tunnels to enable the adjustment of new tunnel parameters and to secure the security of existing large-

diameter shield tunnels. Figure 6 depicts the monitoring section configuration along the existing tunnel. The distance between the essential monitoring portions was 5 m (McS5-13), while the spacing between other monitoring sections was 6 m (McS14-15, McS1-4). A total of 17 monitoring sections were installed along the existing tunnel's longitudinal direction. Figure 6b depicts the overall structure of the monitoring locations with existing tunnel sections. Two monitoring sites, denoted A and B, were installed on both rails of the up-and-down lines. It was also critical to monitor the increased load and joint displacement generated by the installation of new tunnels in existing tunnels. Given the current condition, the 6th and 12th monitoring portions of the existing large-diameter shield were chosen as the strain gauge, reinforcement gauge, and joint gauge monitoring sections. Figure 7 depicts the positions of each monitoring element's measuring points. As can be seen in Figure 7 the reinforcement gauge was arranged symmetrically inside and outside, and the pressure box was arranged one segment apart. The installation direction of the bolt force and joint displacement was circumferential. Because the substructure was prefabricated, the surface seam gauge could not be installed in the substructure. The earth pressure is arranged on the outside of the segment and is poured with the segment. The steel bar stress and concrete stress meter were arranged on the inside and outside of the segment and were poured with the segment. The bolt stress measuring point was first arranged on the bolt and then installed on the segment. The site installation and layout are shown in Figure 8. Detailed information on the sensors is shown in Table 1.

**Table 1.** Detailed information on the sensors.

| Number | Instrument | Model | Producer | Accuracy | Range |
|---|---|---|---|---|---|
| 1 | Joint angle meter | JTM-V7000A | JTM | ≤1.5%F·S | 0~50 mm |
| 2 | Concrete strain gauge | JTM-V5000 | JTM | ≤1.5%F·S | −800 με~1200 με |
| 3 | Steel stress gauge | JTM-V1000 | JTM | ≤2.0%F·S | −200 MPa~100 MPa |
| 4 | Earth pressure transducer | JTM-V2000D | JTM | ≤2.0%F·S | 0~4 MPa |

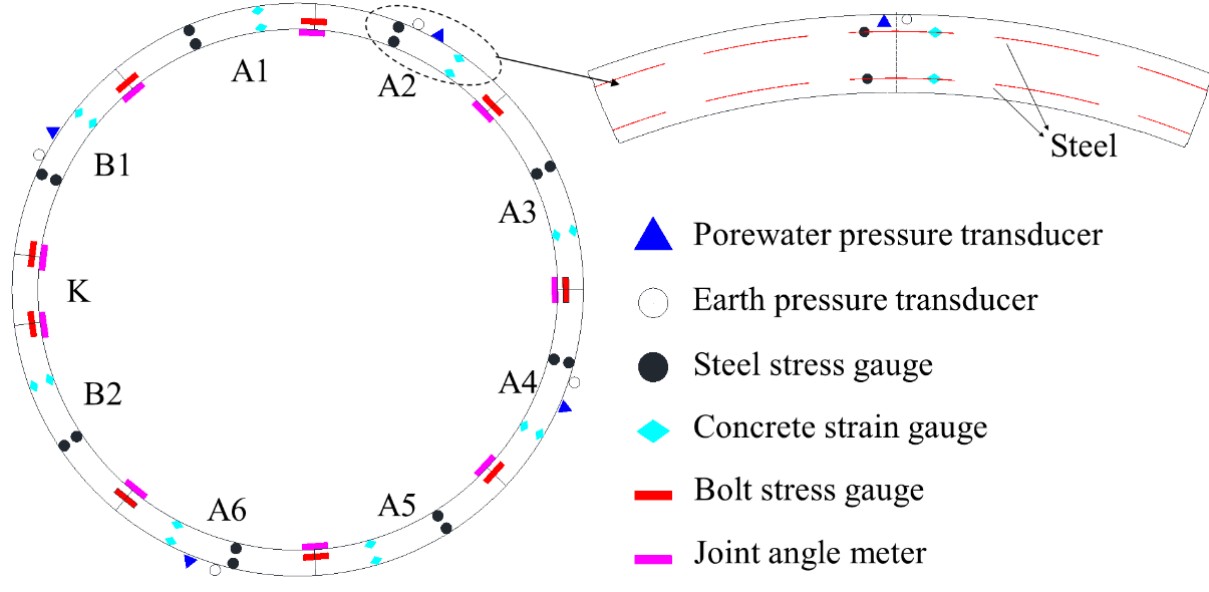

**Figure 7.** Schematic layout of component measurement points.

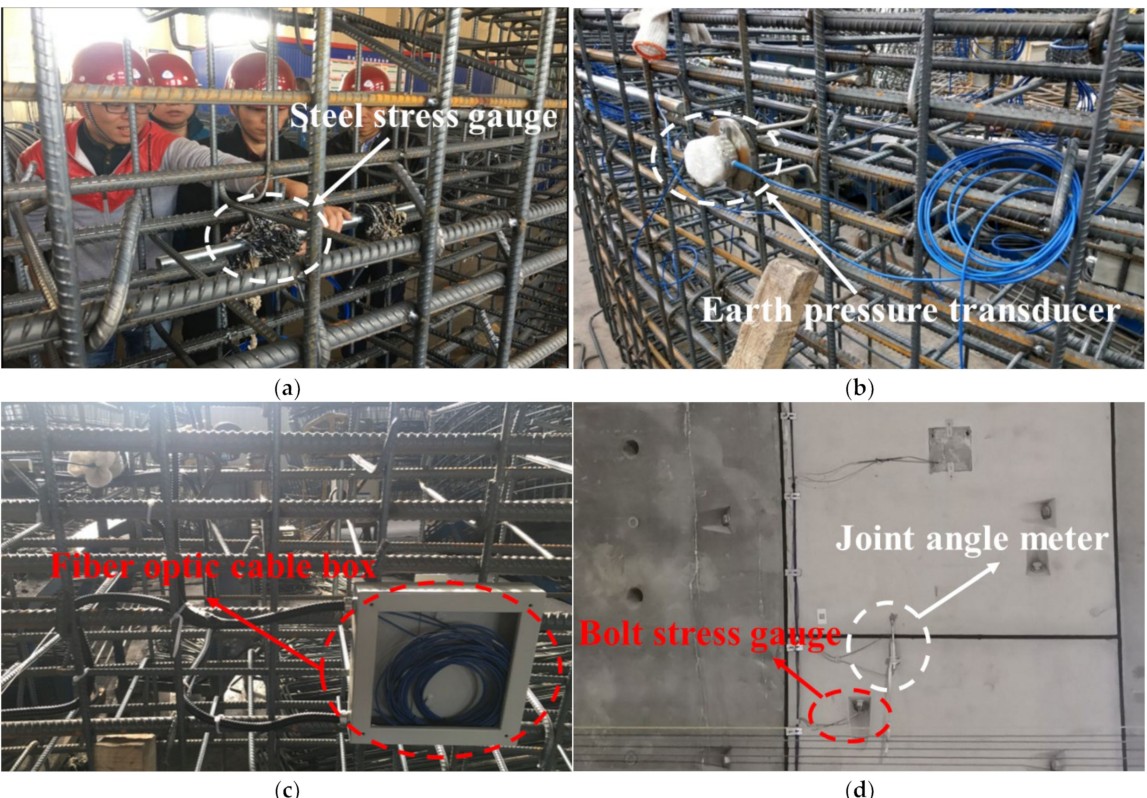

**Figure 8.** The on-site installation and layout. (**a**) Diagram of reinforcement gauge installation. (**b**) Installation diagram of the earth pressure box. (**c**) Fiber optic cable box. (**d**) Appearance of part.

### 3.2. Construction Process of Horseshoe-Shaped Twin Tunnels

The right tunnel was built underneath the existing large-diameter tunnel on 1 December, and the left tunnel was built underneath the existing large-diameter tunnel on 20 December. The left tunnel face lies roughly 10 m behind the right tunnel face in this section of the project, and the distance from the top of each tunnel to the step increases from 4 m to 5 m.

## 4. Monitoring Data Analysis

### 4.1. Settlement Development of the Existing Large-Diameter Tunnel

The deformation law of the shield tunnel of Line 12 between the Jimen Bridge and Dazhongsi interval under the existing Beijing-Zhang High-Speed Railway of Qinghua Park was analyzed, and the vertical deformation ephemeral curve of the existing tunnel is shown in Figure 9. According to the construction sequence and control measures of Line 12's undercrossing construction, the settlement ephemeris curve of the existing tunnel was divided into the following six stages. MSC11-B and MSC12-B, which were positioned on the rail plate, had similar settlement tendencies. Depending on the monitoring data, six stages of vertical deformation development in the existing large-diameter shield tunnel were identified in the location of the new tunnel face and building measures, as shown in Figure 9, and the trend of settlement changes in each stage was categorized as follows:

Stage I: Far-field excavation (>2 D)

During this phase, the maximum settlement of the existing large-diameter tunnel was 1.2 mm. This was due to the new right tunnel face being excavated to the edge of the existing large-diameter shield tunnel by 12 m.

Stage II: Forepoling reinforcement

During this phase, the existing tunnel experienced approximately 2.1 mm of uplift due to the construction aid of overrunning deep hole grouting, which was caused by the use of deep hole grouting technology in sand, and the pebble strata could effectively squeeze

the soil and water body, enhance the strength and stiffness of the soil, and form a certain strength arch shell in the grouting area to reduce the ground deformation caused by tunnel excavation. It could also achieve the effect of grouting jacking to reserve a certain space for the deformation and settlement caused by subsequent excavation. The design's grouting pressure was 0.2 to 0.8 MPa, and the actual grouting pressure was 1.4 MPa.

Stage III: Right line underpass construction

The settlement of the existing tunnel was approximately 3.4 mm at this stage. The excavation and unloading of the new tunnel resulted in a significant increase in the settlement of the existing large-diameter tunnel during tunnel construction. To reduce the deformation caused by the unloading of the strata, the excavation area could be reduced to minimize the disturbance of the strata.

Stage IV: Deep hole grouting in the left line and post-wall grouting in the right line

In the fourth stage, a bulge of approximately 1.2 mm appeared in the existing tunnel. According to the previous construction progress, it is known that at this time, the left line tunnel is undergoing deep hole grouting of the soil ahead as well as backfill grouting (pressurized grouting) behind the right line tunnel's lining to reduce the deformation caused by the unloading of the ground after the excavation of the right line tunnel.

Stage V: Left line under crossing construction

During this phase, the existing tunnel settlement was approximately 1.7 mm. In the excavation phase of the upper steps of the new left line tunnel, the excavation of the soil below the existing tunnel led to certain unloading below, which in turn increased the displacement of the existing large-diameter tunnel in the direction of the new tunnel.

Stage VI: Grouting and stabilization behind the left line wall

When the newly built left line tunnel's lower steps led down across the existing tunnel, the new tunnel excavation on the existing large-diameter tunnel deformation was also not significant. The second lining of the right tunnel increased the stiffness of the support structure, so the deformation gradually tended to be a stable state. The new left tunnel was then grouted behind the wall, and the existing tunnel bulged slightly.

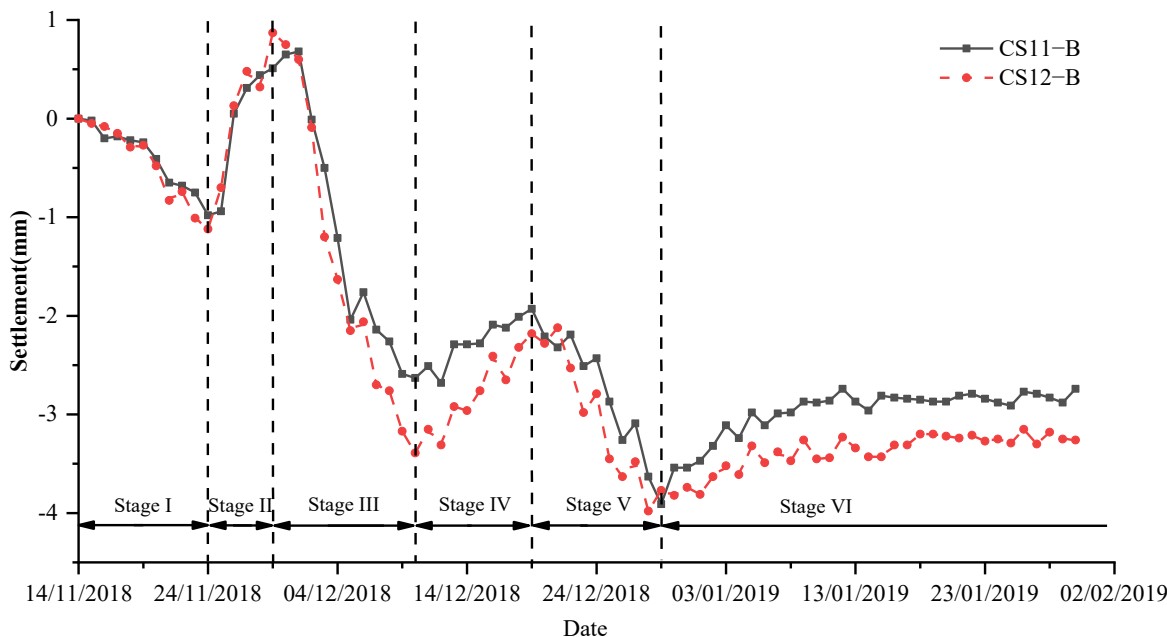

**Figure 9.** Subsidence evolution of the existing large-diameter tunnel during the undercrossing twin tunnels.

*4.2. Settlement Profiles of the Existing Large-Diameter Tunnel*

Figure 10 shows the measured settlement profile of Line 12 after crossing the existing large-diameter tunnel. As illustrated in the picture, the settling of the existing large-

diameter tunnel induced by the crossing of Line 12's right line was parallel with the centerline of the new right line tunnel. The maximum settlement was approximately 3.14 mm. The existing tunnel's subsidence following the second crossing of the left tunnel was asymmetrical about the left tunnel centerline. The maximum settlement position deviated 3.0–6.0 m from the midline of Line 12's left tunnel. This might have been due to the soil disturbance created by the initial crossing as well as the grouting reinforcing measures. Section 4.5 analyzes the reasons for the deviation of the maximum settlement position. The maximal settlement induced by the left line tunnel crossing was 2.16 mm, far less than the settlement generated by the first passing. The settlement part of the present structure was shaped like a "V" after the first crossing but like a "W" after the second crossing. The greatest settlements in the existing tunnel occurred where the new tunnel connected the existing structure, and special attention should be paid to crossing construction in the future.

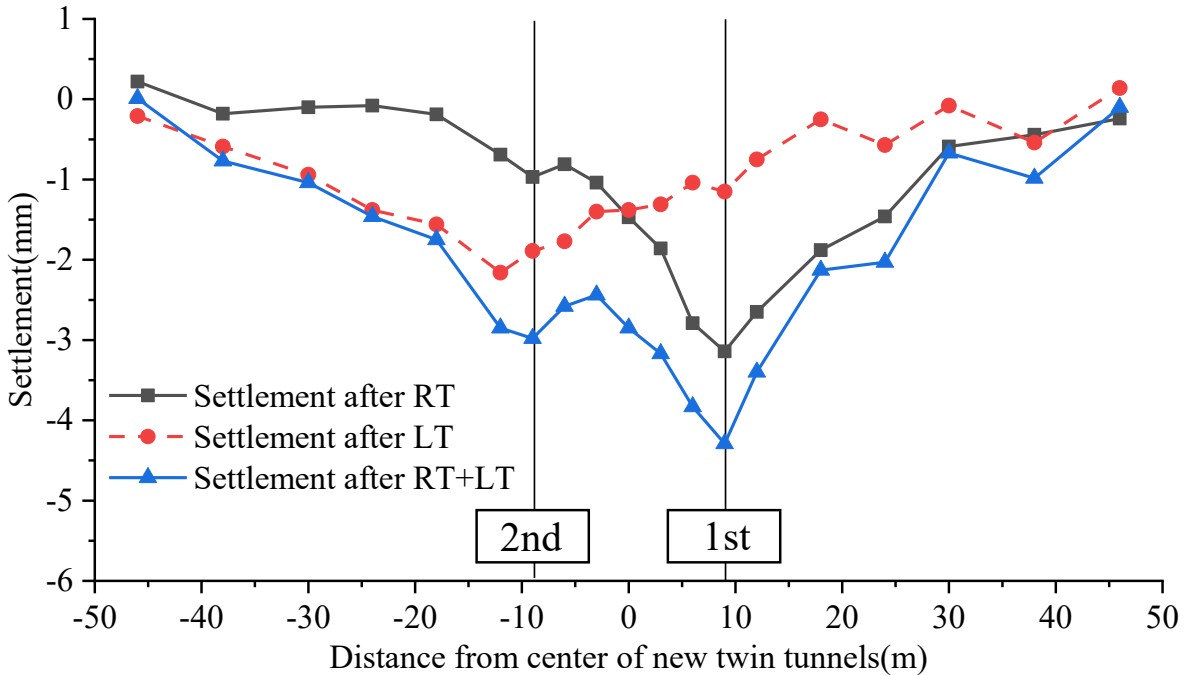

**Figure 10.** Subsidence profiles of the existing large-diameter shield tunnel.

### 4.3. Induced Hoop Stress of the Existing Large-Diameter Tunnel

This section analyzes the circumferential stress changes in the existing large-diameter shield tunnel during the new tunnel's construction. The circumferential stress changes induced by the tunnel construction are shown in Figure 11. The monitoring section is located in ring 785 of the existing large-diameter shield tunnel (see Figure 7 MSC12 monitoring section). As shown in Figure 7, only the data for the eight strain gauges numbered 1, 2, 3, 4, 5, 6, 7, and 9 are shown in Figure 11 due to the failure of strain gauge number 8. The circumferential stress was equal to the pipe segment's modulus of elasticity (34.5 GPa) multiplied by the strain. In Figure 11, the tensile stress inside the tube sheet is considered positive stress, while the compressive stress is considered negative stress. The phases depicted in the diagram correspond to the building phases described below:

1. When the length between both the new tunnel face excavation and the edge of the existing tunnel exceeded 12 m, the circumferential tension at the strain gauge position remained almost unchanged.

2. The circumferential stresses at all strain gauge locations fell when the tunnel face of the right tunnel met the edge of the existing tunnel, except for the circumferential stress at numbered location 7, which increased. According to the preceding description, the stress decrease could be attributed to the existing tunnel's secondary grouting.

3. When the right tunnel crossed the existing tunnel, the circumferential stresses at strain gauges 1 and 5 decreased dramatically, and the minimum extra stress was approximately 3.5 MPa at the tunnel's L-springline, whereas the circumferential stress at strain gauge 7 increased dramatically to approximately 0.4 MPa. This was most likely due to soil disturbance induced by the excavation of the new right tunnel, which resulted in partial unloading and hence an increase in the tensile stress at the arch's bottom.

4. Subsequently, in the left line tunnel crossing the existing tunnel (Figure 11d,e), the stress at number 7 decreased and then increased, probably as a result of the combined effect of deep hole grouting and excavation unloading. The progressive increase in circumferential stress at the other strain gauge locations could be attributed to the excavation's slow upward expansion of the soil arch above the initial tunnel, resulting in a gradual increase in compressive stress. The stress at each site, however, was significantly less than the compressive and tensile strength of C50 concrete, and therefore the pipe piece was safe.

5. With the completion of the new left line tunnel crossing for some time, the deformation and stress of the original tunnel gradually recovered due to the secondary grouting of the left line and part of the recovery capacity of the soil. After the new left line tunnel crossing has been completed for some time, the displacement and strain of the existing tunnel progressively recover due to secondary grouting of the left line and part of the soil's recovery capacity.

*4.4. Joint Openings of the Existing Large-Diameter Tunnel*

The opening width of the segment joints is critical for the tunnel's structural safety, and an excessive opening size might result in groundwater leaking. The opening width of the joint was measured at each longitudinal joint position in this work, as illustrated in Figure 7.

Figure 12 depicts the variations in the longitudinal joint displacement of the existing large-diameter tunnel during the twin tunnels' advance. The monitoring cross-section can be found at the 785th ring of the existing large-diameter shield tunnel (see Figure 6). In Figure 12, the opening displacements of the segment joints are considered positive, whereas compressive displacements are considered negative.

Depending on the observations, the displacement evolution tendencies of 1, 2, 3, 4, 5, and 6 on the existing large-diameter tunnel were similar. However, the magnitudes were different. Each stage depicted in the image corresponds to the stages of construction indicated in Section 4.1 (see Figure 9).

In Stage 3, the longitudinal joint displacement of the existing large-diameter tunnel rapidly increased during the right tunnel of subway Line 12's advancement. The displacement of the longitudinal joints was approximately 0.2~1.2 mm in this stage.

Backfill grouting behind the lining and multiple additional grouting (the first nonpressure backfill grouting and the final four pressure grouting) were performed in Stage 4 to decrease distortion of the existing tunnel. The left line tunnel grouted and reinforced the soil beneath the existing tunnel, which could improve soil strength and stiffness, reduce displacement induced by construction, and achieve the grout jacking effect. The injection pressure was designed to be 0.5 MPa, but it was 1.2 MPa. The displacement of the longitudinal joints was significantly reduced in this stage compared with the first stage, with the maximum change in displacement of the longitudinal joints in the existing tunnel being 0.7 mm (B1–A1 longitudinal joint displacement), which may be due to the joint action of the secondary grouting and the overgrowing range.

The longitudinal seam displacement grew rapidly in Stage 5 and was roughly 2.9 mm after the conclusion of the left tunnel crossing construction (28 December 2018), indicating that the existing tunnel's longitudinal seam displacement was more susceptible to the new tunnel underpass development.

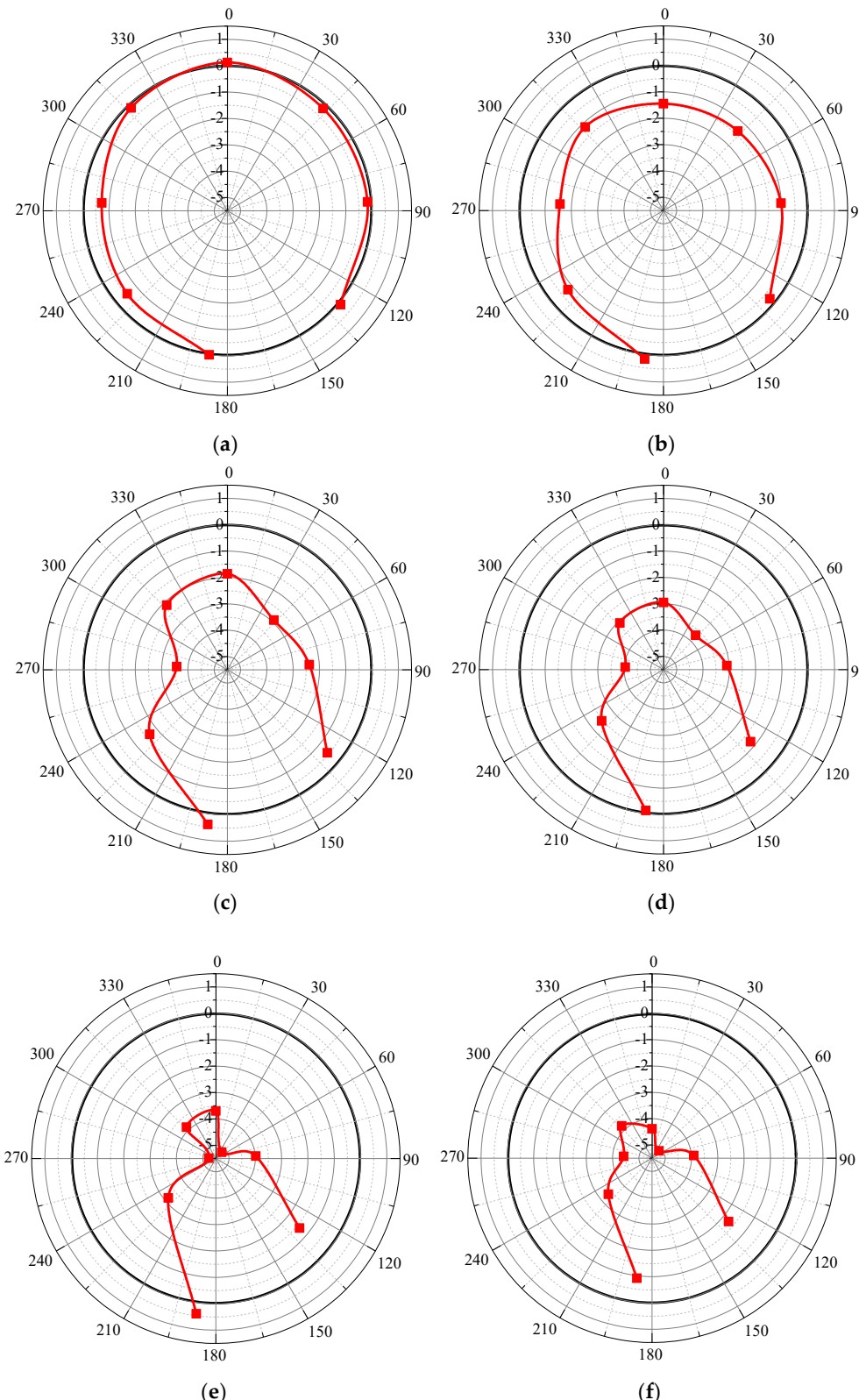

**Figure 11.** Induced hoop stress of the existing tunnel in the transverse section. (**a**) Far-field excavation; (**b**) Forepoling reinforcement; (**c**) Right line underpass construction; (**d**) Deep hole grouting in the left line and post-wall grouting in the right line; (**e**) Left line under crossing construction; (**f**) Grouting and stabilization behind the left line wall.

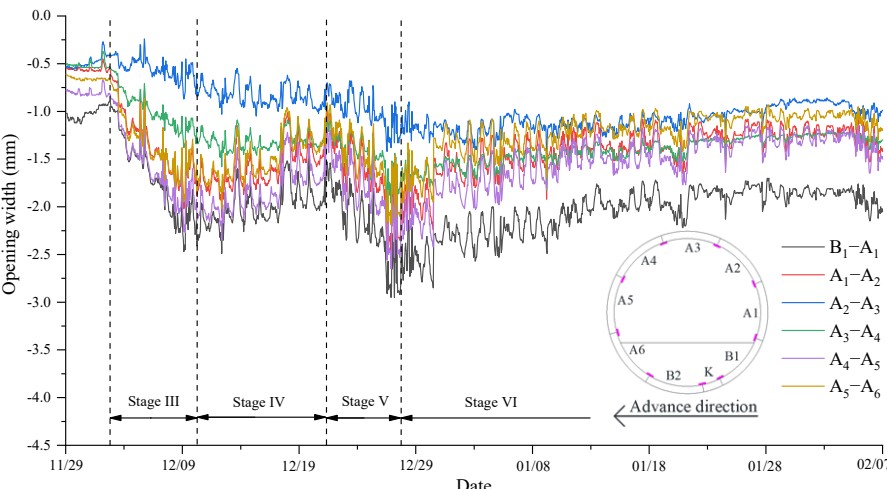

**Figure 12.** The opening width of the segment joint versus time at CS12.

In Stage 6, secondary grouting of the left line was carried out, and the secondary lining structure was gradually poured, strengthening the support structure's integrity and rigidity. As a result, the displacement of the longitudinal joints in the existing tunnel first decreased rapidly and then decreased slowly and steadily. From the above four stages experienced by the longitudinal seam displacement in the 785-ring tunnel, the settlement occurring in the first stage accounted for most of the total settlement, so this stage needs the attention of engineers.

Figure 12 depicts the displacement curve of the existing tunnel's longitudinal joints in ring 776. The development process of the longitudinal seam displacement caused by the new tunnel excavation below could also be separated into several phases: right line crossing and left line crossing (stage 1), the grouting recovery phase (stage 2), the left line crossing phase (stage 3), and the restoration of stability phase (stage 4). They correspond to each construction process of the new cut-and-cover two-lane tunnel. The evolution of the longitudinal seam displacement in the 776-loop tunnel is similar to that in the 785-loop tunnel (Figure 13). We will not elaborate too much on this here.

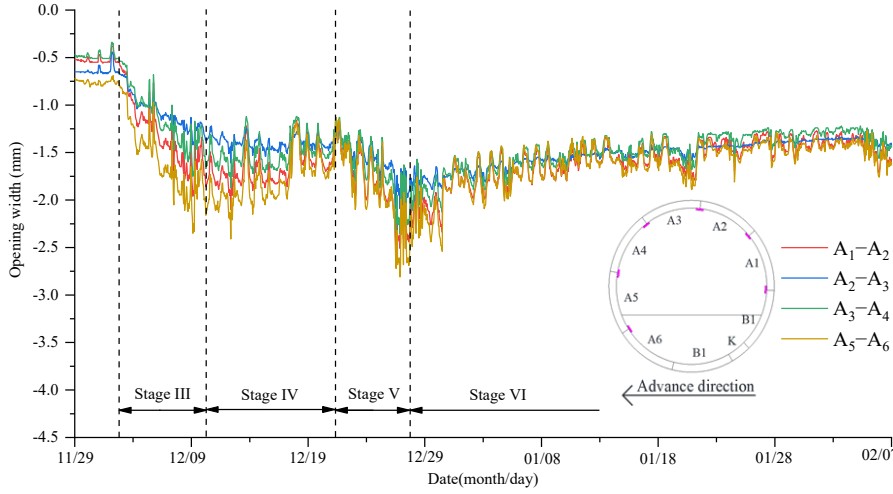

**Figure 13.** The opening width of the segment joint versus time at CS6.

### 4.5. Interactions between Twin Tunnels and Their Influence on the above Tunnel

The Gaussian distribution curve was first developed to represent the land subsidence profile, and it has since been confirmed by numerous researchers in a variety of practical engineering and laboratory tests [25].

Furthermore, the existing large-diameter shield tunnel is a flexible construction, so the Peck equation can be utilized to characterize the settling trough generated by tunnel excavation [26]. However, in a complex metropolitan setting, the construction of double tunnels might have a greater impact on neighboring existing tunnels, and the ultimate settlement is not a simple superposition of two Gaussian distribution curves. The interaction between new twin tunnels is also the primary source of the asymmetric settlement profile. The settlement trough induced by the first tunnel excavation may be calculated using the Peck equation. Figure 14 depicts separate settlement profiles generated by Line 12's twin tunnel excavation.

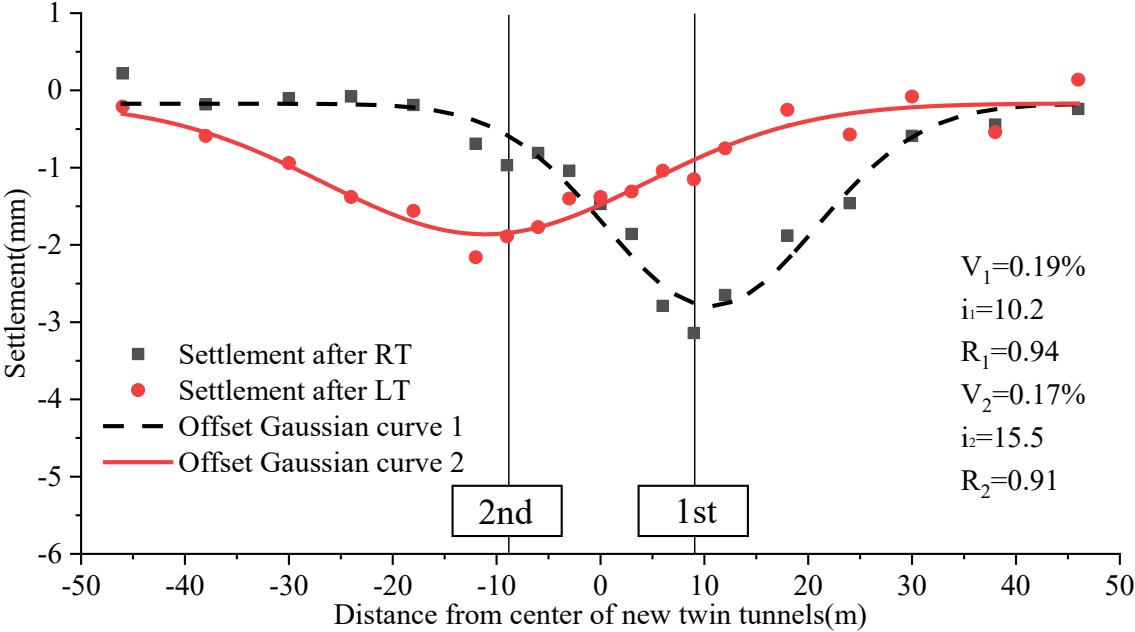

**Figure 14.** Measured and fitted settlement profiles.

As shown in Figure 14, the volume loss due to the second tunnel's construction (0.14%) was found to be less than the volume loss due to the first tunnel's construction (0.19%). The width of the settlement tank induced by the second tunnel, on the other hand, was slightly greater than that induced by the first tunnel.

A large number of examples investigated the changes in the sedimenting tank generated by the first and second tunnel excavations. However, in this case, it is also observed that there was a difference between the settlement trough induced by the construction of the first tunnel and that of the second tunnel. The largest settlement induced by the second tunnel construction occurred at a distance from the center of the new twin tunnel rather than on its central line. The reason for this may be because the soil in the middle part was reinforced before the construction of both the left and right lines of Line 12, resulting in multiple reinforcements that greatly changed the stiffness of the soil.

### 4.6. Earth Pressure Evolution and Stress Characteristics of the Existing Large-Diameter Tunnel

Figure 15 depicts the progression of ground pressure and the mechanical responses of the existing large-diameter tunnel. The monitoring portion is located on the existing large-diameter tunnel's 785th ring (see Figure 6).

The soil pressure and steel bar stress increased slightly but not much in the first and second stages. When the first tunnel began to penetrate, the soil pressure and steel stress at A1, A3, and A5 increased sharply. The maximum ground pressure was 53 kPa in the tunnel vault position, while the maximum steel stress was 1.3 MPa in the tunnel's left arch. The growth rate was drastically slowed in the fourth stage. The effect of the second tunnel underpass on the existing large-diameter tunnel was considerably weakened in comparison with the effect of the first tunnel underpass because the monitoring section

was above the first new tunnel. The earth pressure and stress on each segment were steady once the tunnel passed through.

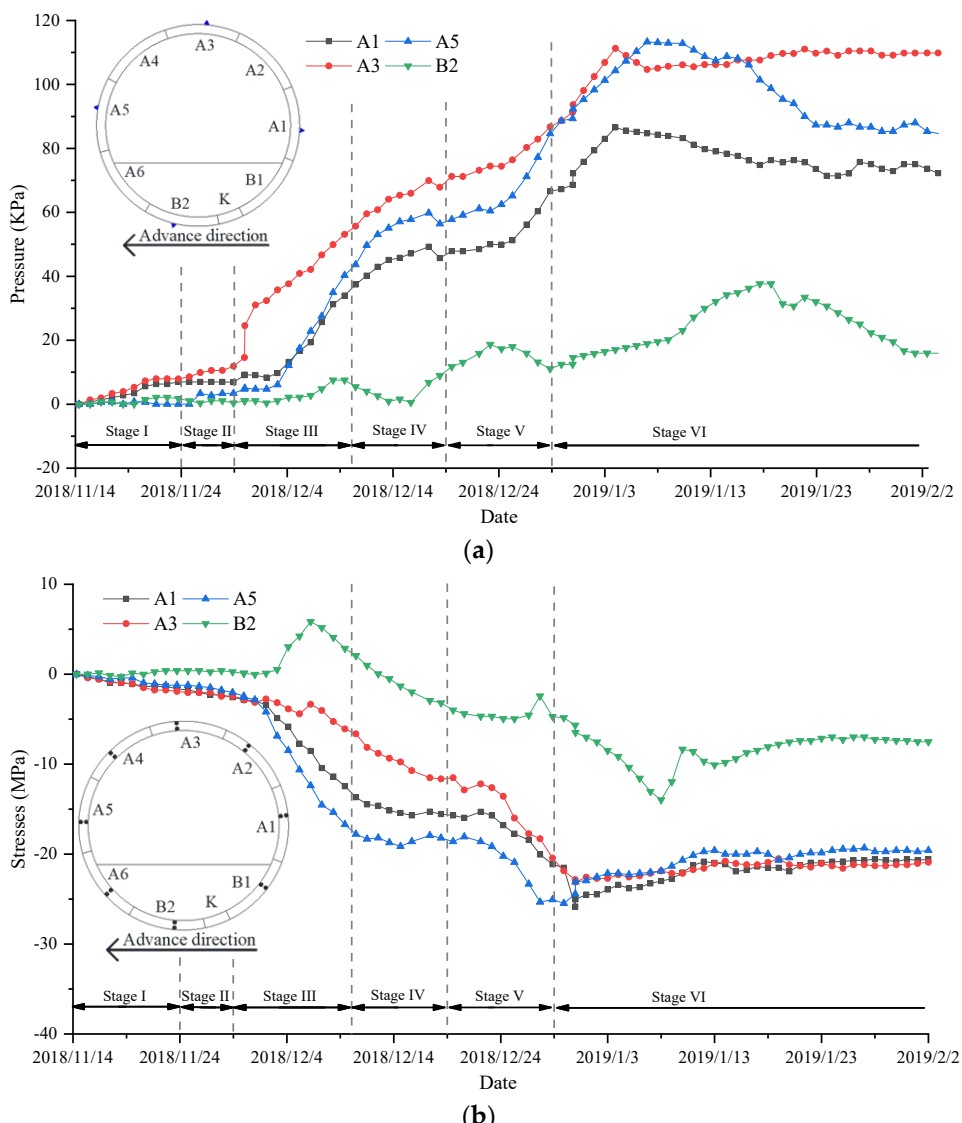

**Figure 15.** Earth pressure evolution and mechanical responses of the existing large-diameter tunnel: (**a**) earth pressure and (**b**) reinforcement stress.

## 5. Discussion

This study investigated the mechanical responses of the existing large-diameter tunnel in Beijing, China caused by a close distance by horseshoe-shaped undercrossing twin tunnels in gravel. Based on the field monitoring data, the mechanical responses of the twin tunnels undercrossing the existing large diameter tunnel were analyzed comprehensively. In this paper, it is proposed that the deformation law of the existing large-diameter shield tunnel can provide qualitative analysis for the design and construction of the underpass project and grasp the deformation situation first to understand the weak links in the project.

In this paper, it is proposed that the deformation law of an existing large-diameter shield tunnel can provide qualitative analysis for the design and preliminary construction of underpass engineering to grasp the deformation situation in advance and understand the weak links in the engineering. This case study provides a valuable reference for similar projects in the future. However, different construction methods of new tunnels will lead to different deformation and internal force laws in existing tunnels. The deformation

and internal force laws summarized in this paper are still not detailed enough. In future research, the deformation and internal force of the existing tunnel can be summarized in detail according to the method of new tunnels, which can better provide qualitative analysis for future projects.

## 6. Conclusions

This paper focuses on the mechanical responses of existing large-diameter tunnels induced by a close-distance shallow tunneling method (STM) undercrossing in sand and gravel strata in Beijing, China. The following are some of the significant conclusions from this study:

1.  The subsidence evolution of the existing large-diameter tunnel during the undercrossing twin tunnels showed six stages, including (1) the advanced settlement (Stage 1) caused by the construction disturbance in far-field excavation (>2 D), (2) the heave induced by the effect of the grouting pressure, (3) the vertical unloading caused by volume loss at the stratum during tunnel passage having a significant impact on the second settling, (4) the rise induced by the left tunnel deep hole grouting and right tunnel wall back grouting, (5) the third settlement resulting once again in state disturbance during the left tunnel passing, and (6) the settlement development of the existing large-diameter shield tunnel slowly decreasing and then gradually stabilizing in this stage. In conclusion, grouting measures have a great influence on the existing tunnel deformation.

2.  The subsidence profile of the existing large-diameter shield tunnel caused by left line tunnel crossing is asymmetric relative to the central line of the second tunnel. The maximum settlement point was approximately 3–6 m away from the center line of the left line tunnel of subway Line 12, which may indeed be related to the soil disturbance of the first tunnel's construction and grouting reinforcement measures. The subsidence and volume loss due to the second tunnel's construction was found to be less than the volume loss due to the first tunnel's construction. This may be due to the reinforcement of the middle part of the left and right lines before the construction of the left and right lines or Line 12, which caused many reinforcements and greatly changed the stiffness of the soil. The final subsidence profile of the existing large-diameter shield tunnel was a "W" shape after the second tunnel passed through.

3.  When the right line tunnel passed through the existing tunnel, the circumferential stress changed dramatically, with a size of roughly 0.4 MPa at the left arch line of the existing tunnel. The reason for this could be that the undercrossing architecture caused some unloading, which increased the tensile stress at the arch bottom.

4.  The opening and compression of the existing large-diameter shield tunnel's joints were inextricably linked to the new tunnel's excavation process. Undercrossing construction will lead to the opening of the joints, and thus the effectiveness of the waterproof function at the joint is significantly reduced. However, the undercrossing effect can be weakened by secondary pressure grouting for the existing large-diameter shield tunnel before excavation.

**Author Contributions:** J.L. and Q.F.: conceptualization, methodology, project administration, and writing the original draft; X.L., J.D., G.W., and J.W.: data collection, plotting curves and analysis, and review. All authors have read and agreed to the published version of the manuscript.

**Funding:** The authors gratefully acknowledge the Key Project of High-Speed Rail Joint Fund of the National Natural Science Foundation of China grant number U1934210.

**Institutional Review Board Statement:** Not applicable.

**Informed Consent Statement:** Not applicable.

**Data Availability Statement:** The data used to support the findings of this study are available from the corresponding author upon request.

**Conflicts of Interest:** The authors declare no conflict of interest.

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
