# Peer review of "Mechanical Behaviors of Existing Large-Diameter Tunnel Induced by Horseshoe-Shaped Undercrossing Twin Tunnels in Gravel"

_applsci, doi:10.3390/app12147344_

Round 1

Reviewer 1 Report

The novelty of this paper is significant enough to publish on “Applied Sciences”. Due to some issues that need improvement before publication, therefore my decision is acceptance with minor revisions.

Here are my comments for improving manuscript:

1.      Introduction:

·   The introduction provides background information and set the context. Therefore, to make introduction clearer, please consider to state research questions and research objectives.

2.      Literature Review:

· Literature Review section plays an important role in the research, therefore please add this section in this manuscript.

3.      Monitoring data analysis: The results are quite good, however there is a lack of discussions. Please consider to discuss thoroughly research limitations and practical implications.

4.      Conclusion: conclusion will be improved by emphasizing the research contributions and future work. However, there is lack of recommendations for future research. Please add these.

5.      References:

·    Some references are too old (over ten years). Please consider to update new ISI articles to improve this section.

Reviewer 2 Report

This is a very interesting case study on the Subway excavation below the existing Qinghuayuan tunnel in Beijing.

However, it has lack of a convenient *article structure*. "*Material and methods*, *Results*, as well as *Discussion*" were missed in the manuscript. In my opinion, if the authors could rearrange its structure, it would be more convenient for a research paper (and not be confused with a “Report”). Besides, I would expect more information on the sensors used as accuracy, range, model, producer, etc. Please review the cohesion of the text for a research paper and English writing. For example, avoid using *past tense* in a sentence and in the next one utilizing *present tense*.

I recommend the paper to go through “Major Revisions”.

Round 2

Reviewer 1 Report

The novelty of this paper is significant to publish in the Journal. Thus, my decision is an acceptance for a publication.

Reviewer 2 Report

After the required revisions, the paper is ready to publish.